# Single molecule analyses reveal dynamics of *Salmonella* translocated effector proteins in host cell endomembranes

Vera Göser[1], Nathalie Sander[1], Marc Schulte[1,4], Felix Scharte[1,4], Rico Franzkoch[1,2,4], Viktoria Liss [2], Olympia E. Psathaki[2,3] & Michael Hensel [1,3] ✉

The facultative intracellular pathogen *Salmonella enterica* remodels the host endosomal system for survival and proliferation inside host cells. *Salmonella* resides within the *Salmonella*-containing vacuole (SCV) and by *Salmonella*-induced fusions of host endomembranes, the SCV is connected with extensive tubular structures termed *Salmonella*-induced filaments (SIF). The intracellular lifestyle of *Salmonella* critically depends on effector proteins translocated into host cells. A subset of effectors is associated with, or integral in SCV and SIF membranes. How effectors reach their subcellular destination, and how they interact with endomembranes remodeled by *Salmonella* remains to be determined. We deployed self-labeling enzyme tags to label translocated effectors in living host cells, and analyzed their single molecule dynamics. Translocated effectors diffuse in membranes of SIF with mobility comparable to membrane-integral host proteins in endomembranes. Dynamics differ between various effectors investigated and is dependent on membrane architecture of SIF. In the early infection, host endosomal vesicles are associated with *Salmonella* effectors. Effector-positive vesicles continuously fuse with SCV and SIF membranes, providing a route of effector delivery by translocation, interaction with endosomal vesicles, and ultimately fusion with the continuum of SCV/SIF membranes. This mechanism controls membrane deformation and vesicular fusion to generate the specific intracellular niche for bacterial survival and proliferation.

Various intracellular pathogens are confined to membrane-bound compartments. Within these organelles, pathogens are able to adopt specific intracellular lifestyles. Biogenesis of specialized pathogen-containing vacuoles depends on recruitment of subsets of host cell endosomes in order to establish nutritional supply, and to evade the host immune defense. For this purpose, pathogens translocate, by different secretion systems, specific effector proteins that manipulate the host cell endosomal system[1].

*Salmonella enterica* is a Gram-negative, foodborne bacterial pathogen, causing diseases ranging from severe typhoid fever to self-limiting gastrointestinal infections in various hosts. *S. enterica* serovar Typhimurium (STM) is commonly used to investigate the

[1]Abt. Mikrobiologie, Universität Osnabrück, Osnabrück, Germany. [2]iBiOs – Integrated Bioimaging Facility Osnabrück, Osnabrück, Germany. [3]CellNanOs – Center of Cellular Nanoanalytics Osnabrück, Osnabrück, Germany. [4]These authors contributed equally: Marc Schulte, Felix Scharte, Rico Franzkoch. ✉e-mail: Michael.Hensel@uni-osnabrueck.de

intracellular lifestyle of *Salmonella*. After invasion or phagocytic uptake, STM initiates a complex intracellular lifestyle enabling survival and proliferation within host cells. STM resides in a membrane-bound compartment termed *Salmonella*-containing vacuole (SCV), which acquires late endosomal markers, but does not mature to a bactericidal compartment[2]. Characteristic of infected cells is the formation of tubular structures connected to the SCV. Such *Salmonella*-induced tubules (SIT) comprise various tubular structures composed of recruited host endomembranes of various organellar origin[3]. The best characterized SIT are *Salmonella*-induced filaments (SIF), marked by lysosomal membrane glycoproteins such as LAMP1[4]. SIF are highly dynamic in the initial phase of intracellular lifestyle. If fully developed, SIF comprise double membranes built up during development where initial SIF are single-membrane tubular compartments (leading SIF), which over time, convert into double-membrane (trailing SIF) tubular structures[5]. The molecular mechanisms of these pathogen-driven events of vesicle fusion and membrane deformation remain to be determined.

SIF formation and systemic virulence of STM are dependent on functions of genes within *Salmonella* pathogenicity island 2 (SPI2). SPI2 encodes a type III secretion system (T3SS) which enables the translocation of various effector proteins inside the host cell[6]. Mutant strains deficient in SPI2-T3SS are highly attenuated in systemic virulence in the mouse model of systemic infection, and show reduced intracellular replication in cell-based models[7,8]. We recently reported that SIF biogenesis supports intracellular lifestyle by bypassing nutritional restriction in SCV-SIF continuum by recruiting nutrients from the host endosomal system and is therefore crucial for bacterial proliferation and survival[9].

Despite the large number of effector proteins translocated by the SPI2-T3SS, only a subset of these manipulates the host endosomal system and induces vesicle fusion for SCV and SIF biogenesis. These are SifA, SseF, SseG, PipB2, SseJ and SopD2[6]. The most severe phenotype is mediated by SifA, as mutant strains defective in *sifA* fail to induce SIF and show loss of SCV integrity leading to bacterial release into host cytosol, and attenuation in intracellular proliferation and systemic virulence[10]. SseF, SseG, and PipB2 contribute to SIF formation, as mutant strains lacking the effectors show aberrant SIF biogenesis. Infection with *sseF*-deficient STM leads to the formation of only single-membrane SIF, and infection with *pipB2*-deficient strains results in the induction of enlarged bulky SIF[5,11].

A subsets of SPI2-T3SS effectors is recruited to *Salmonella*-modified membranes (SMM) after translocation that is prominently associated with membranes of the SCV and SIF network[12]. This subcellular localization appears to be prerequisite for effector and host protein interactions that mediate vesicle fusion and SIF elongation[13]. However, the molecular mechanisms of effector targeting to SMM are poorly understood. In STM-infected cells, a dynamic extension of SIF network was observed, raising the question how SPI2-T3SS effector integrate into SMM. For example, highly hydrophobic effectors like SseF appear exclusively associated with SIF membranes[14,15]. As T3SS translocation delivers effector proteins into host cell cytosol, specific mechanisms of targeting and integration into host endosomal membranes are required, and we here applied novel imaging approaches to unravel these mechanisms.

We recently established an imaging approach utilizing self-labeling enzyme (SLE) tags fused to STM effector proteins to enable super-resolution microscopy (SRM), and single-molecule imaging of effector dynamics in living cells[16]. Here we applied these approaches to investigate the delivery of SPI2-T3SS effectors to SMM, their localization, and dynamics in SMM. This study provides new insights in the delivery mechanism of effector proteins to the SCV-SIF continuum.

## Results

### Continuous interactions of SCV, SIF, and host cell endosomal compartments

We analyzed interactions of intracellular STM with the host cell endosomal system. Pulse/chase experiments with fluorochrome-conjugated gold nanoparticles (nanogold) allowed to label the lumen of endosomes[17]. A subset of these endosomes was in contact with dynamic SIF and events of fusion between nanogold-labeled endosomes and SIF were detected (Supplementary Movie 1). Due to the transient nature, fusions between host cell endosomes and membranes of SCV or SIF were rarely determined, and Supplementary Movie 1 shows a representative event. In contrast to other fluid tracers that become rapidly diluted after a fusion between endosomes and SIF, the aggregation of nanogold led to the formation of distinct foci that were readily detectable after fusion events. We next performed ultrastructural analyses of STM-infected cells with markers for vesicular fusions (Supplementary Fig. 1). Decoration of endosomal membranes with LAMP1-miniSOG lead to 3,3'-diaminobenzidine (DAB) conversion in endosomal lumen and deposition of DAB polymers in SIF lumen (Supplementary Fig. 1a). Pulse/chase of STM-infected cells with nanogold BSA-rhodamine labeled vesicular compartments associated and fusing with SCV and SIF (Supplementary Fig. 1b).

Interactions of endosomal compartments with SIF were frequently observed in TEM analyses of infected host cells (Supplementary Fig. 2a). Using live-cell correlative light and electron microscopy (CLEM), infected cells were imaged during the formation of dynamically extending SIF (Fig. 1, Supplementary Fig. 2b–i). Correlation identified LAMP1-positive tubular vesicles in connection with SCV harboring STM. The investigated cell showed double-membrane (dm) SIF distal to the SCV (Fig. 1e), or in connection to SCV (Fig. 1f, g). In few occasions, the contact of single-membrane (sm) vesicles of spherical appearance with dm SIF was observed (Fig. 1h, i, j). Although the temporal resolution of our live-cell imaging (LCI) approach did not allow us to distinguish fusion from fission events for the vesicle, our data would be in line with a fusion of a host cell endosome with a dm SIF and release of luminal content in the outer lumen of SIF.

### Distribution of SPI2-T3SS effector proteins on SCV and SIF membranes

The previous data revealed the fusogenic properties of the SCV/SIF continuum, and indicated how the SIF network is dynamically expanding. Prior work demonstrated that the formation of the SIF network is dependent on translocated SPI2-T3SS effector proteins, and that subsets of these effector proteins are closely associated with SIF membranes[15,18,19]. Thus, we next followed the distribution of PipB2 as representative SPI2-T3SS effector protein over the course of STM infection (Fig. 2a). In the early phase (4 h p.i.), the signal intensity for PipB2 immunostaining was low, and most of the signals were associated with small spherical vesicles. At 8 h p.i., a SIF network was developed and PipB2 signals were mostly associated with SIF and SCV membranes. At the end of observation, i.e. 16 h p.i., PipB2 signal was strongly increased and was almost exclusively colocalized to membranes of SIF and SCV. A similar subcellular distribution over time of infection was observed for other membrane-associated SPI2-T3SS effectors such as SseF and SseJ (Supplementary Fig. 3).

To investigate if translocated effector proteins are present in membranes of endosomal compartments prior to integration of these membranes into the SCV/SIF continuum, we applied immunogold labeling for TEM analyses. A triple HA-tagged allele of *sseF* was used as representative membrane-integral SPI2-T3SS effector protein. SseF-3xHA was synthesized, translocated, and subcellular localized as observed for SseF-HA (Supplementary Fig. 4). For optimal preservation of endosomal membranes and epitopes we applied the Tokuyasu technique[20] for immunolabeling on ultrathin hydrated sections. In

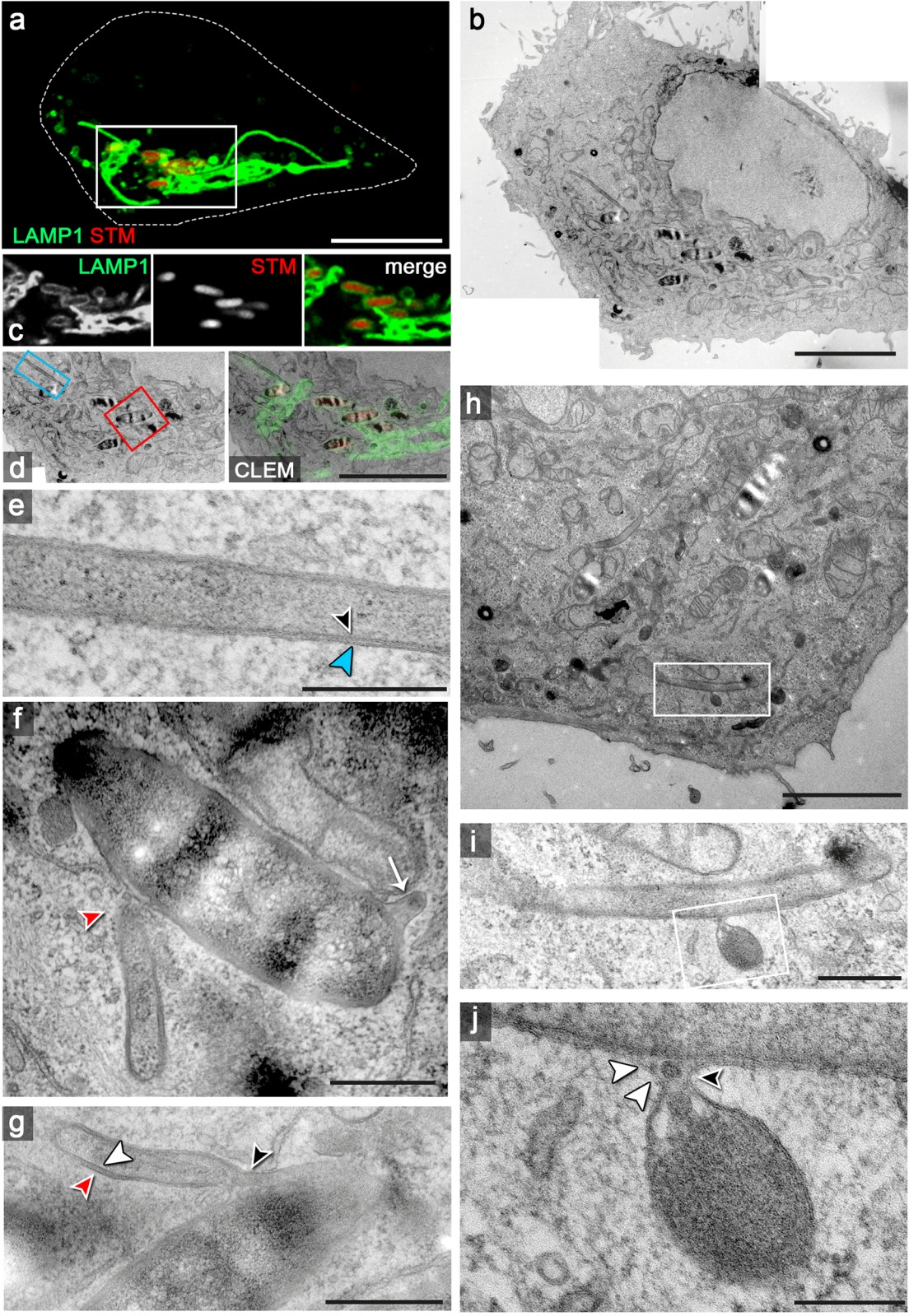

**Fig. 1 | Interactions of host cell endosomal membranes in STM-infected cells.**
HeLa cells stably expressing LAMP1-GFP (green) were infected with STM WT
expressing mCherry (red) and CLSM was performed (**a**, **c**) to identify SIF-positive
cells showing dynamic extension of SIF networks. Cells were fixed 7 h p.i., coordi-
nates registered and samples were processed for TEM (**b**). Correlation of CLSM and
TEM modalities allowed identification of STM in SCV and extending SIF tubules (**d**).
Regions of interest are indicated by boxes and details show a double-membrane
(dm) SIF tubule distal to SCV (blue, **e**), and in connection with the SCV (**f**, **g**). The
white box in **h** indicates an event of vesicle interaction with a dm SIF, and details are
shown in higher magnification (**i**, **j**). Micrographs show events representative from
four independent experiments, and further events are shown in Supplementary
Fig. 1 and Supplementary Fig. 2. Arrowheads indicate interactions with double-
membrane compartments, while single-membrane tubules are indicated by arrows.
Scale bars: 10 μm, 5 μm (**b**, **c**), 3 μm (**h**), 500 nm (**e**, **f**, **i**), 300 nm (**g**), 200 nm (**j**).

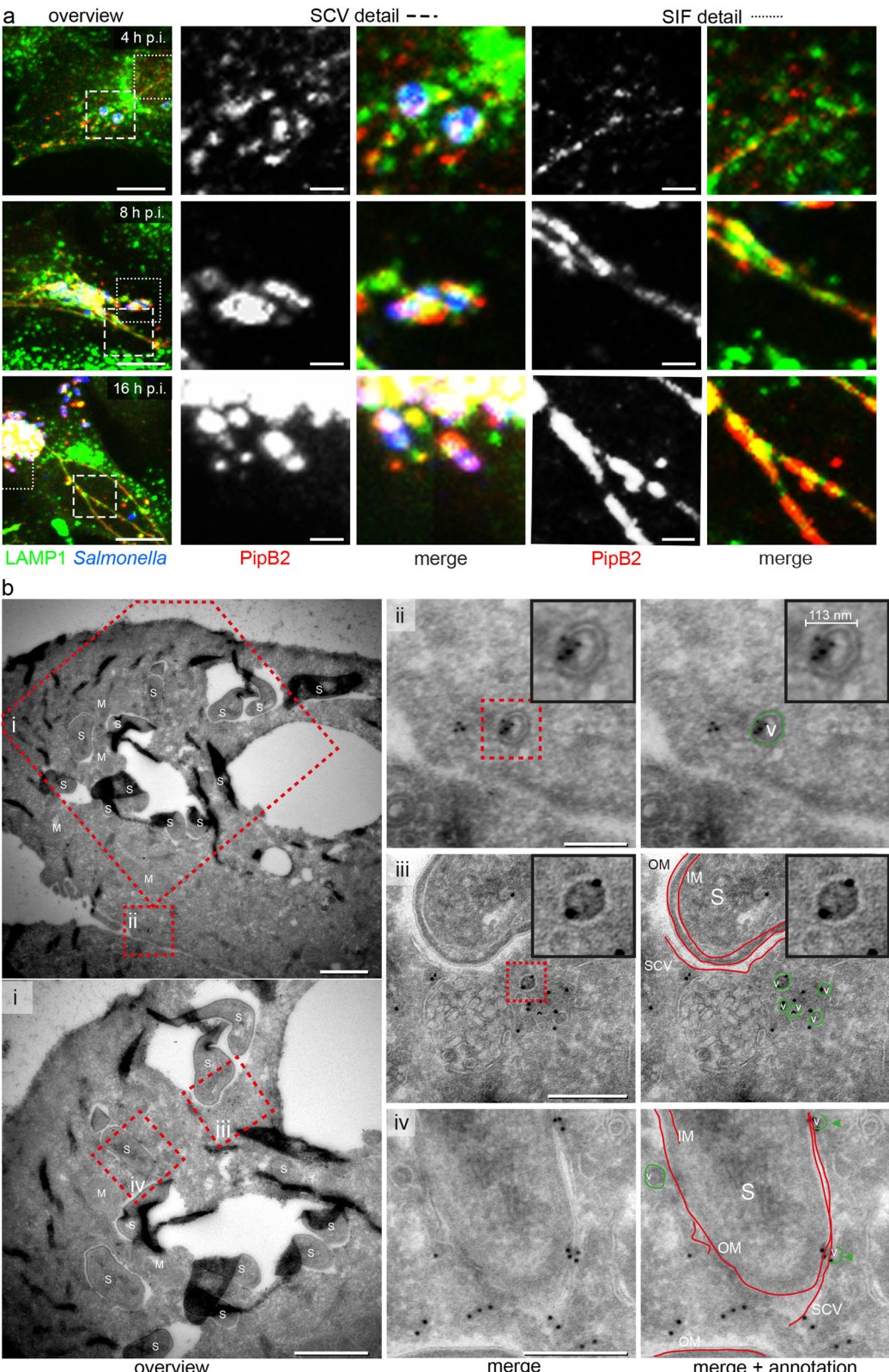

**a**

overview SCV detail – – – SIF detail ·······

LAMP1 *Salmonella* PipB2 merge PipB2 merge

**b**

overview merge merge + annotation

infected HeLa cells, immunogold-labeled SseF-3xHA was associated with SCV membranes (Fig. 2). We also detected labeling for SseF associated with spherical membrane compartments distal to the SCV. In ultrathin sections, such signal could result from cross-sections of small spherical vesicles, or of extended tubular compartments such as SIF. To distinguish these forms, consecutive ultrathin sections were inspected, indicating labeling a single section rather than in compartments extended through several sections (Supplementary Fig. 4).

These ultrastructural observations support a model that effector proteins associate with and integrate in host cell endosomal membranes prior to incorporation into the SCV/SIF continuum.

**Models for SPI2-T3SS effector targeting to endomembranes**

It is not known how hydrophobic effector proteins insert into host cell endomembranes. We built several hypotheses for the route of SPI2-T3SS effector proteins from translocation to their final

**Fig. 2 | Kinetics of distribution of SPI2-T3SS effector proteins and vesicular localization of translocated SseF. a** Distribution of translocated effector proteins over the course of infection. HeLa cells stably expressing LAMP1-GFP (HeLa LAMP1-GFP) were infected with STM WT expressing *pipB2*::M45. At various time points after infection, cells were fixed and immunolabeled for STM (blue) and effector proteins (red). Details of SCV and SIF are shown. Micrographs show events representative from three independent experiments, and further events and time points are shown in Supplementary Fig. 3. Scale bars: 10 and 2 μm in overview and details, respectively. **b** Vesicular localization of translocated SseF revealed by immunogold EM. HeLa LAMP1-GFP cells were infected with STM WT expressing *sseF*::3xHA and fixed 8 h p.i. The samples were processed for immunogold labeling for HA-tagged SseF. Details of overviews (**b**, **i**) of SseF immunogold-labeled sections are shown in **ii–iv**. (**ii**) A subset of triple HA-tagged SseF immunogold labeling is associated with the outer and inner side of spherical vesicular membranes. See also color high-lighted vesicle structure in green on the left. Inserts strongly clarify localization of immunogold label inside the vesicle and on the vesicular membrane. (**iii**) Triple HA-tagged SseF immunogold labeling is also found on endomembranes mostly in close proximity to vesicles. Color marking in green for vesicles and red for SCV, inner (IM) and outer (OM) bacterial membrane is highlighting the distribution of gold labeling on membrane structures. Inserts strongly clarify localization of immunogold label on the vesicular membrane. **iv**) The majority of triple HA-tagged SseF immunogold labeling is distributed on endomembranes, specifically on membranes closely associated with the SCV and directly on the SCV. See also color marking in red indicating for SCV membrane, IM and OM. Micrographs show events representative from three independent experiments, and further sections are shown in Supplementary Fig. 4. Scale bars: 1 μm in overviews (**b**, **i**); 250 nm in **ii–iv**.

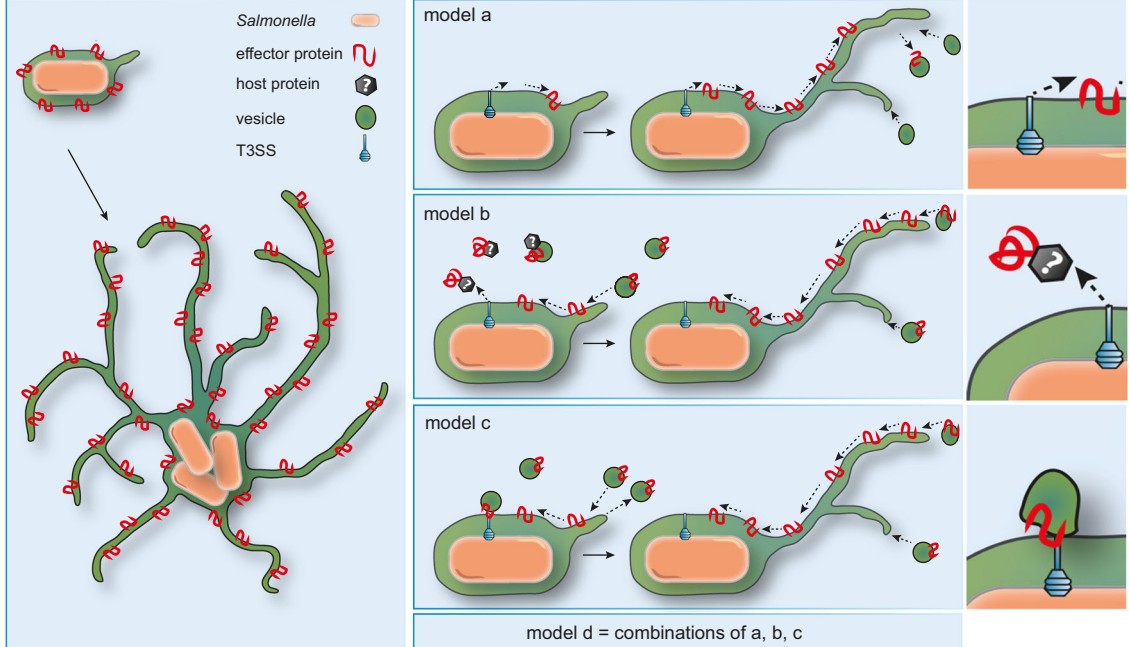

**Fig. 3 | Models for targeting of SPI2-T3SS effector proteins to host cell endosomal membranes.** Model **a**: Effector proteins are directly inserted into the SCV membrane after translocation and diffuse from the SCV to the periphery of SIF. Model **b**: Effector proteins are translocated into the cytosol, chaperoned, and inserted into endomembranes by unknown host factors. Effector proteins are delivered by fusion of host endosomal vesicles with SIF and SCV. Model **c**: Endomembranes are recruited to SCV and T3SS where effector proteins are directly delivered and inserted into endomembranes. After fusion of endosomal vesicles effector proteins are delivered. In addition to fusion of endosomal vesicles to SCV and SIF, budding of effector-positive vesicles from the SCV/SIF continuum may be considered. Combinations of models **a–c** may be considered.

destination (Fig. 3). In model **a**, effector proteins are directly integrated into SCV membranes after translocation. In model **b**, effector proteins are translocated into the host cell cytosol, and a fast interaction with unknown bacterial or host cell proteins enables insertion into host endomembranes. In model C, direct delivery of effector proteins into host vesicular membranes is mediated by the SPI2-T3SS itself, and no cytosolic effector intermediates are present. In model **a**, peripheral distribution of effector proteins is mediated by tubulation of SCV membranes containing effector proteins. In models **b** and **c**, effector proteins are first inserted into endosomal membranes that subsequently fuse with developing SIF. We would also consider combinations of the models, and distinct modes of delivery for different effector proteins. We set out to test these models by applying a recently developed LCI approach for translocated effector proteins on single molecule level[16].

## SPI2-T3SS effector proteins are highly dynamic on SIF membranes

To follow the dynamics of SPI2-T3SS effector proteins on or in SIF membranes, we deployed single-molecule localization and tracking microscopy (TALM)[21]. As host cells, HeLa cells were used that constitutively express LAMP1-monomeric enhanced green fluorescent protein (LAMP1-GFP) to allow visualization of SCV and SIF. Host cells were infected with STM mutant strains deficient in genes for specific effectors. The strains harbored plasmids encoding effector proteins fused to HaloTag, a SLE tag, and infected cells were labeled with HaloTag ligand coupled to the fluorescent dye tetramethylrhodamine (HTL-TMR). As previously shown[16], the effector proteins SseF, SifA, and PipB2 fused to HaloTag can be localized in infected host cells 8 h p.i. and a complete colocalization with LAMP1-GFP-positive SCV and SIF membranes was observed (Fig. 4a, Supplementary Fig. 5, Supplementary Movies 2, 4, 5).

We tracked the movement of the key STM effector proteins SseF, SifA, and PipB2 fused to HaloTag on *Salmonella*-modified membranes. By analyzing comprehensive data sets of single-molecule trajectories, the mobility of effector proteins using pooled trajectories resulting in a two-dimensional diffusion coefficient (DC), extracted from mean square displacements (MSD), was demonstrated[16]. As control, the host membrane protein LAMP1 was used and visualized after transient transfection of HeLa LAMP1-GFP cells for expression of LAMP1-

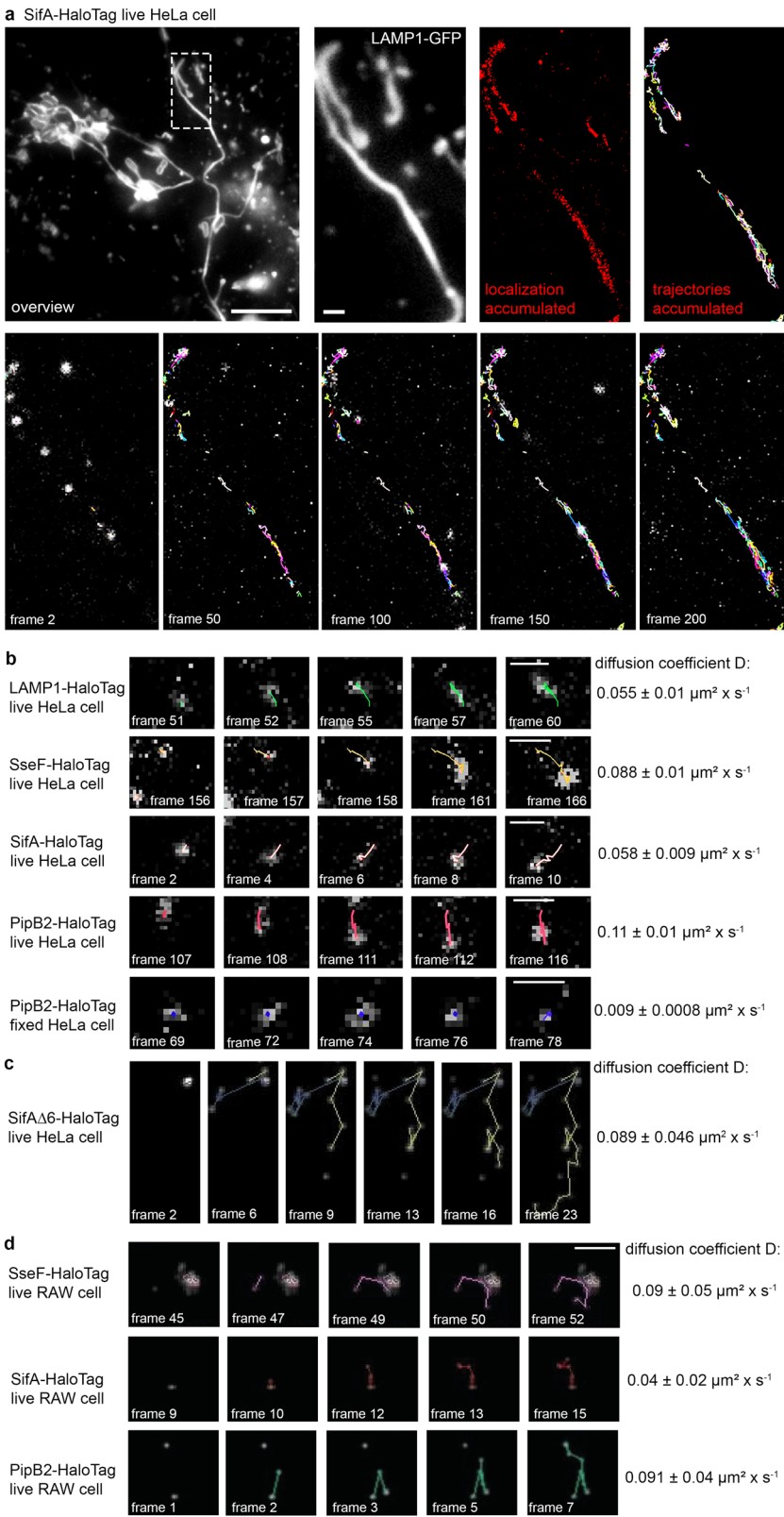

HaloTag (Supplementary Movie 3). For non-moving particles, tracking of PipB2-HaloTag on SIF tubules was performed in fixed host cells (Supplementary Movie 6).

The DC of fixed PipB2-HaloTag was quantified as $0.009 \pm 0.0008\ \mu m^2 \times s^{-1}$. For LAMP1-HaloTag, a DC of $0.055 \pm 0.01\ \mu m^2 \times s^{-1}$ was determined. The effector proteins SifA, SseF, and PipB2 fused to HaloTag varied in their mobility with DC of $0.058 \pm 0.009$, $0.088 \pm 0.01$ and $0.11 \pm 0.01\ \mu m^2 \times s^{-1}$, respectively (Fig. 4b, and Supplementary Movies 2, 4, 5). These values are in line with data in our previous report describing the technique[16]. In all cases, the trajectories developed bidirectional, without preferential movement of molecules towards SCV-proximal or SCV-peripheral portions of SIF.

The SPI2-T3SS effector proteins investigated here all are associated with host cell endosomal membranes after translocation. For

**Fig. 4 | Single molecule localization and tracking of STM SPI2-T3SS effector proteins on double-membrane SIF.** HeLa cells stably expressing LAMP1-GFP were infected with STM *sifA* mutant strain expressing SifA-HaloTag with a multiplicity of infection (MOI) of 75. Following incubation for 7 h under standard cell culture conditions, LCI was performed. Labeling reactions were performed directly before imaging, using HTL-TMR with a final concentration of 20 nM for 15 min at 37 °C. **a** Shown are representative SRM images acquired using 15% laser power at the focal plane, rendered from single molecule localizations (SML) and tracking (SMT) within 200 consecutive frames. Selected frames (frame rate: 32 frames per s) of the TMR signal, localization, and tracking are presented (also showing elapsed trajectories). **b** Selected frames of trajectories from a single molecule of indicated effector-HaloTag fusions. Using at least 2800 pooled trajectories for proteins in at least 20 infected cells in three biological replicates recorded under the same conditions, the diffusion coefficient D was calculated using the Jaqaman algorithm. The indicated error represents the calculated error of the resulting slope (with 95% confidence bounds). A sequence of 200 frames for SifA-HaloTag is shown in Supplementary Movie 2. **c** SML and SMT analyses of mutant SifA-HaloTag. STM translocating a SifA allele with deletion of aa 331-336 (SifAΔ6) were used to infect host cells as in **a**) and analyses were performed as for WT SifA. **d** Murine macrophage-like RAW264.7 cells stably expressing LAMP1-GFP were activated by γ-Interferon and infected with STM strains grown to stationary phase at MOI 50, and SML and SMT analyses were performed as for **a**). Selected frames of trajectories from a single molecule of indicated effector-HaloTag fusions in RAW264.7 cells. Scale bars: 10 and 1 μm in overviews and details, respectively.

several effectors, interacting host proteins are known, such as SKIP and PLEKHM1 for SifA[22,23]. For SifA, the endomembrane targeting is due to prenylation of a C-terminal CxxC motif[24]. We investigated if the ablation of membrane binding of SifA alters dynamics in host cells. A mutant SifA allele lacking the CxxC motif (SifAΔ6-HaloTag) was translocated by intracellular STM and showed association with SIF membranes. This allowed SMT, and mean DC of $0.089\,\mu m^2 \times s^{-1}$ (±0.046) was determined (Fig. 4c, Supplementary Fig. 6). The DC of SifAΔ6-HaloTag was about twice the motility of SifA-HaloTag, indicating that lack of prenylation severely increases diffusion rate.

We also mutated coiled-coil domains proposed to mediate membrane association of PipB2, SseJ, or SopD2[25]. The mutant alleles of these effectors also showed altered motility on SIF membranes. Further studies will reveal the role of these domains in interaction with endomembranes and effects on mobility.

To test the broader physiological relevance of effector dynamics, we used an infection model with the commonly used murine macrophage cell line RAW264.7. RAW264.7 permanently transfected for LAMP1-GFP expression were used to phagocytose stationary phase STM. After adaptation to intracellular conditions, STM initiated SPI2-T3SS translocation resulting in endosomal remodeling and SIF formation. Translocated effector proteins fused to HaloTag were analyzed SMT as shown for SifA-HaloTag (Fig. 4d, Supplementary Fig. 7). The movement of SseF, PipB2 and SifA on SIF membranes was analyzed and DC were determined (Fig. 4d, Supplementary Fig. 7). The DC for SseF, SifA, and PipB2 fused to HaloTag were similar to those determined in HeLa cells. As observed for HeLa cells, in RAW264.7 the DC of SifA was lower than DC of SseF or PipB2.

Infection with STM mutant strain ΔsseF leads to increased formation of sm SIF which are smaller in diameter and volume. Fully developed SIF in STM WT-infected cells are predominantly dm SIF[5,11]. We analyzed SPI2-T3SS effector mobility on sm SIF to investigate potential effects of SIF architecture on the distribution and diffusion of effector proteins. HeLa LAMP1-GFP cells with STM ΔsifA ΔsseF strain expressing *sifA*::HaloTag and dynamics of SifA-HaloTag molecules on sm SIF were analyzed (Fig. 5a, b, Supplementary Movies 7, 8). The reduced diameters of sm SIF were verified by intensity profile analyses of accumulated SifA-HaloTag trajectories on SIF induced by STM WT and ΔsseF strains (Fig. 5d). When calculating DCs for LAMP1-HaloTag and SifA-HaloTag on sm SIF induced in cells infected by STM ΔsseF, a reduction of mobility with DC values of $0.028 \pm 0.008$ and $0.035 \pm 0.004\,\mu m^2 \times s^{-1}$, respectively was observed (Fig. 5c).

Taken together, these single-molecule analyses demonstrate that SPI2-T3SS effector proteins are highly dynamic on SIF. PipB2 showed distinct higher mobility in comparison to host membrane-integral protein LAMP1. The mobility of SifA and LAMP1 was reduced on sm SIF in comparison to dm SIF.

### SPI2-T3SS effectors accumulate on leading SIF during transition to trailing SIF

After succeeding in imaging effectors on sm SIF, we analyzed the transition of leading to trailing SIF. Thinner leading SIF consists of single-membrane tubules and the connected trailing SIF consist of double-membrane structures, i.e. fully developed SIF. It was proposed that this transition is facilitated by a lateral extension of membranes of leading SIF, engulfment of host cell cytosol and cytoskeletal filaments, and finally membrane fusion to form double-membrane trailing SIF[5]. To image the transition from leading to trailing SIF, translocated SseF-HaloTag was analyzed by LCI at 6 h p.i. In Fig. 5e and Supplementary Movie 9, a thin SIF with a weak GFP signal was imaged, and a wider trailing SIF with strong GFP signal was developing alongside the thinner structure. By collecting and localizing all signals of SseF-HaloTag between the frames of the growing LAMP1-positive SIF tubule, increased concentration of effector proteins on the leading SIF before transition to trailing SIF became apparent. This was also shown for SifA-HaloTag and PipB2-HaloTag (Supplementary Fig. 8a, b, Supplementary Movies 10, 11). These effector proteins were also localized at leading SIF when transforming to trailing SIF, however the local concentration was less pronounced.

These findings indicate that SPI2-T3SS effector proteins are already present on leading SIF, and in particular SseF appears to be involved in the transformation to trailing dm SIF, as an accumulation of effector protein can be detected directly before transition.

### Effector proteins target endosomal vesicles in the early phase of infection

The presence and concentration of effector proteins on leading SIF suggest a delivery mechanism of effectors to the tips of growing sm SIF tubules. To address the question how SPI2-T3SS effector proteins reach their subcellular destination in an infected host cell, we applied LCI by confocal laser-scanning microscopy (CLSM) of infected HeLa LAMP1-GFP cells at 4 h p.i. In the early stage of infection, the SCV is already formed, while SIF biogenesis initiates. We found that after labeling of effector proteins, also bacteria were heavily stained, indicating large amounts of effector proteins stored in bacteria. By monitoring the HaloTag-fused PipB2, SseF, SseJ, and SteC, localization of effector proteins in a punctate, vesicle-like manner was observed in infected cells. These structures showed most frequently colocalization with LAMP1-GFP signal, but also labeled endomembrane compartments lacking the late endosomal marker were observed (Supplementary Fig. 9).

As prior work indicated the interaction of SseF and SseG[26], and function of both effector proteins is required for formation of dm SIF and efficient intracellular proliferation[5,9], we followed the translocation and potential interactions of SseF and SseG (Supplementary Fig. 10a). Immunolabeling of epitope-tagged SseF and SseG indicated differential targeting to LAMP1-positive endosome. Quantification indicated SseF colocalization with LAMP1 already in the early phase of 4 h p.i., while SseG colocalization with LAMP1 was delayed and at lower levels (Supplementary Fig. 10b). The colocalization of SseF and SseG increased over time and colocalized effector were predominantly located on SIF (Supplementary Fig. 10a). To further analyze SseF and SseG interaction, we applied SMT of SseF-HaloTag in the background of STM strains either lacking SseF or both SseF and SseG (Supplementary Fig. 10c). The diffusion coefficient for SseF-HaloTag was

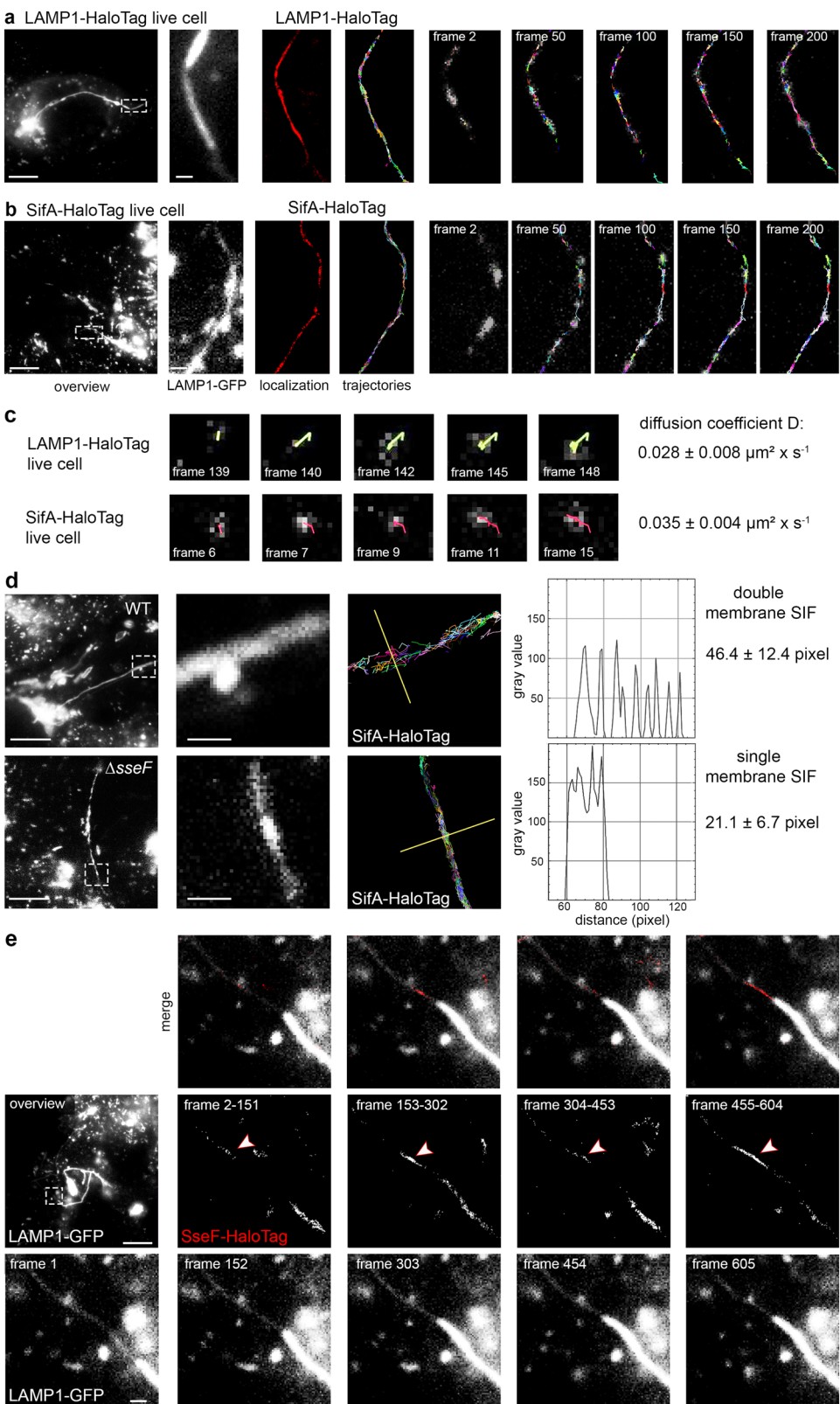

increased in STM Δ*sseF* background compared to STM WT, and slightly lower in background of STM Δ*sseFG*. These data indicate the spatial interaction of SseF and SseG during endosomal remodeling and suggest that the proper balance in the amounts of the translocated effectors affect their interaction.

SifA-HaloTag was not detected decorating vesicles in the early stage of infection. The low level of SifA-HaloTag translocation could hamper visualization. These findings are in line with the observation made by SRM localization with SifA-HaloTag showing the lowest effector concentration, while PipB2-HaloTag showed the highest labeling intensity on SIF tubules. Accordingly, vesicles marked with PipB2-HaloTag could be easily imaged in the early stage of infection, and PipB2 protein was chosen for further analyses. We set out to determine different phenotypes of PipB2-HaloTag localization and therefore studied 100 cells containing PipB2-HaloTag-positive vesicles.

**Fig. 5 | Single molecule localization and tracking of STM SPI2-T3SS effector proteins on single-membrane SIF.** HeLa cells stably expressing LAMP1-GFP were infected with STM *sseF, sifA* mutant strains expressing *sifA*::HaloTag::HA and labeled with HTL-TMR as described above. For visualization of LAMP1-HaloTag, the cells were transfected with LAMP1::HaloTag::HA one day before infection, and infected with STM *sseF* mutant strain. **a, b** Representative SRM images acquired using 15% laser power at the focal plane, rendered from single-molecule localization and tracking within 200 consecutive frames. Selected frames (frame rate: 32 frames per second) of the TMR signal, localization and tracking are presented, (also showing elapsed trajectories). The sequences of 200 frames of SifA-HaloTag and LAMP1-HaloTag are shown in Supplementary Movie 7 and Supplementary Movie 8. **c** Selected frames of trajectories from a single molecule. Using at least 2,800 pooled trajectories for proteins in at least 20 infected cells in three biological replicates recorded under the same conditions, the diffusion coefficient D was calculated applying the Jaqaman algorithm. The indicated error is the calculated error of the resulting slope (with 95% confidence bounds). **d** Intensity profile analysis of SifA-HaloTag trajectories on sm and dm SIF. The intensity profiles of trajectories tracked on SIF were analyzed using the FIJI plot profile tool. SIF from various infected cells were processed and the resulting pixel range of the profile was determined. **e** HeLa cells stably expressing LAMP1-GFP were infected with STM *sseF* mutant strain expressing *sseF*::HaloTag::HA and labeled with HTL-TMR as described above. The transition leading to trailing SIF was imaged with 488 nm laser excitation for 1 frame (frame rate: 32 frames per second) following 561 nm laser excitation for 150 frames in 4 cycles. Shown are representative SRM images acquired using 15% laser power at the focal plane, rendered from SML within each of the 150 consecutive frames. High local concentration of effector protein on leading SIF is indicated by arrowheads. The sequences of 5 frames of LAMP1-GFP are shown in. Scale bars: 10 and 1 μm in overviews and details, respectively.

Of note, at 4 h p.i. effector-positive vesicles were found in a subset of infected cells. We conclude that due to heterogeneity in SPI2 induction, infected cells with bacteria with low levels of effector secretion showed no detectable HaloTag signal. In line with this observation, PipB2-HaloTag fluorescence intensity on SIF at 16 h p.i. also varied between infected cells (Supplementary Fig. 11). In the early phase of infection, distinct phenotypes of PipB2-HaloTag localization can be distinguished for effector-positive vesicular structures. Moreover, the infected cells either showed no SIT, or already developing first SIT structures. The tubular structures were either LAMP1-GFP-positive, or lacking the endosomal marker, and in one population of cells these structures had already acquired PipB2-HaloTag, and others were still lacking the effector (Fig. 6a). At 4 h p.i., in total 62% of the infected cells still did not show SIF formation, yet were positive for vesicles decorated with PipB2-HaloTag. All other cells also displayed vesicles positive for PipB2-HaloTag and already formed SIT (Fig. 6b). To test the characteristics of effector-decorated vesicles in infected cells, we performed tracking analyses of vesicles. Single LAMP1-GFP-positive or PipB2-HaloTag-positive vesicles were tracked in 3D in infected cells. In co-motion tracking analyses, we observed that vesicles positive for LAMP1-GFP and PipB2-HaloTag were tracked in parallel, and the patterns of movement were identical (Fig. 7a, b, Supplementary Movie 12). When studying individual vesicle tracks, as control conditions LAMP1-GFP-decorated vesicles in non-infected cells, either nocodazole-treated or non-treated were tracked (Supplementary Fig. 12, Supplementary Movies 13, 14).

We quantified the mean track displacement length (MTDL) and the mean track speed (MTS) of pooled trajectories. In infected cells, PipB2-HaloTag-marked and LAMP1-GFP-marked vesicles did not differ in MTDL and MTS and therefore showed normal characteristics of vesicle movement. In cells treated with nocodazole, both values were significantly decreased due to the inhibition of vesicle trafficking after microtubule disruption. Interestingly, late endosomal vesicles tracked in non-infected and non-treated cells showed a more rapid movement, and MTDL and MTS values were significantly increased (Fig. 7c). These data demonstrate that SPI2-T3SS effectors are recruited to endomembrane compartments, and as analyzed in detail for PipB2-Halo-Tag, in infected cells in the early phase of infection PipB2-HaloTag-positive vesicles behave similar to LAMP1-positive vesicles.

### PipB2-HaloTag-positive vesicles continuously integrate into the SIF network

After establishing that SPI2-T3SS effector proteins are recruited to vesicles in the early infection during SIF biogenesis, we hypothesized that delivery to SCV and SIF tubules may occur by fusion of endosomal vesicles with membrane-integrated effector proteins. We set out to study PipB2-HaloTag localization from early to late stage of infection and applied long-term LCI of infected cells. Over time, we observed a reduction of effector-decorated vesicles and extension of effector-positive SIF network (Fig. 8a, Supplementary Movie 15). Rendering the PipB2-HaloTag-labeled endomembranes, 29 effector-positive vesicle-like objects were detected 5 h p.i. while only six objects remained at 12 h p.i. Concurrent with the decreasing number of PipB2-HaloTag-decorated objects, PipB2-HaloTag-positive SIF developed (Fig. 8c), suggesting that effector-positive SIF membranes emerge from vesicles. We speculate that STM mutant strains without the ability to induce tubulation of SIF but still possessing a functional SCV should accumulate effector-positive vesicles over time. We used STM deficient in *sifA* and *sseJ*, previously reported to maintain SCV membrane but lacking SIF formation[27]. In infected HeLa LAMP1-GFP cells at 16 h p.i. an accumulation of PipB2-HaloTag-positive vesicles was observed, as well as lack of SIF network formation (Supplementary Fig. 11). These findings indicate that STM SPI2-T3SS effector proteins are integrated into the SCV-SIF continuum via fusion of effector-positive endomembrane compartments.

### Effector recruitment to endomembrane compartments is dependent on vesicle movement

To investigate if presence of effector-decorated vesicles is dependent on dynamics of endosomal compartments, we monitored localization of PipB2-HaloTag over time in infected cells after inhibition of vesicle trafficking. For this purpose, infected cells were treated with nocodazole 2 h p.i., a time point where STM resides in SCV and activates expression of SPI2-T3SS genes[28]. Addition of nocodazole abrogates vesicle movement due to interference with microtubule integrity. We did not observe PipB2-HaloTag-positive vesicles in nocodazole-treated cells, implicating that vesicle movement is required for effector recruitment Fig. 8b, Supplementary Movies 16).

To address reversibility of inhibition, we imaged infected cells for 8 h after removing nocodazole by washing cells twice at 4 h p.i. A slow initiation of vesicle movement was monitored over time. Starting 11 h p.i., the first PipB2-HaloTag-decorated LAMP1-GFP-positive vesicles were imaged (Fig. 8b). Association of effector proteins with endosomal compartments was dependent on vesicle movement on microtubules, but did not require SIF formation.

## Discussion

We previous demonstrated that SPI2-T3SS effector proteins, as well as the host cell protein LAMP1 can be tracked, on single molecule level, on SIF and show a rapid, bidirectional movement[16]. Here were applied SMT to follow the fate translocated effector proteins in infected host cells. The ability of integral membrane proteins to rapidly interchange in SIF membranes has already been demonstrated by FRAP experiments and that revealed recovery of LAMP1 after photobleaching on distal SIF[9]. Using the split-GFP approach, PipB2 recovery on tubules was detected upon photobleaching, indicating a rapid distribution of PipB2[29]. The diffusion coefficients (DC) are similar to values determined for mitochondrial proteins and for cytokine receptors on the plasma membrane[30,31].

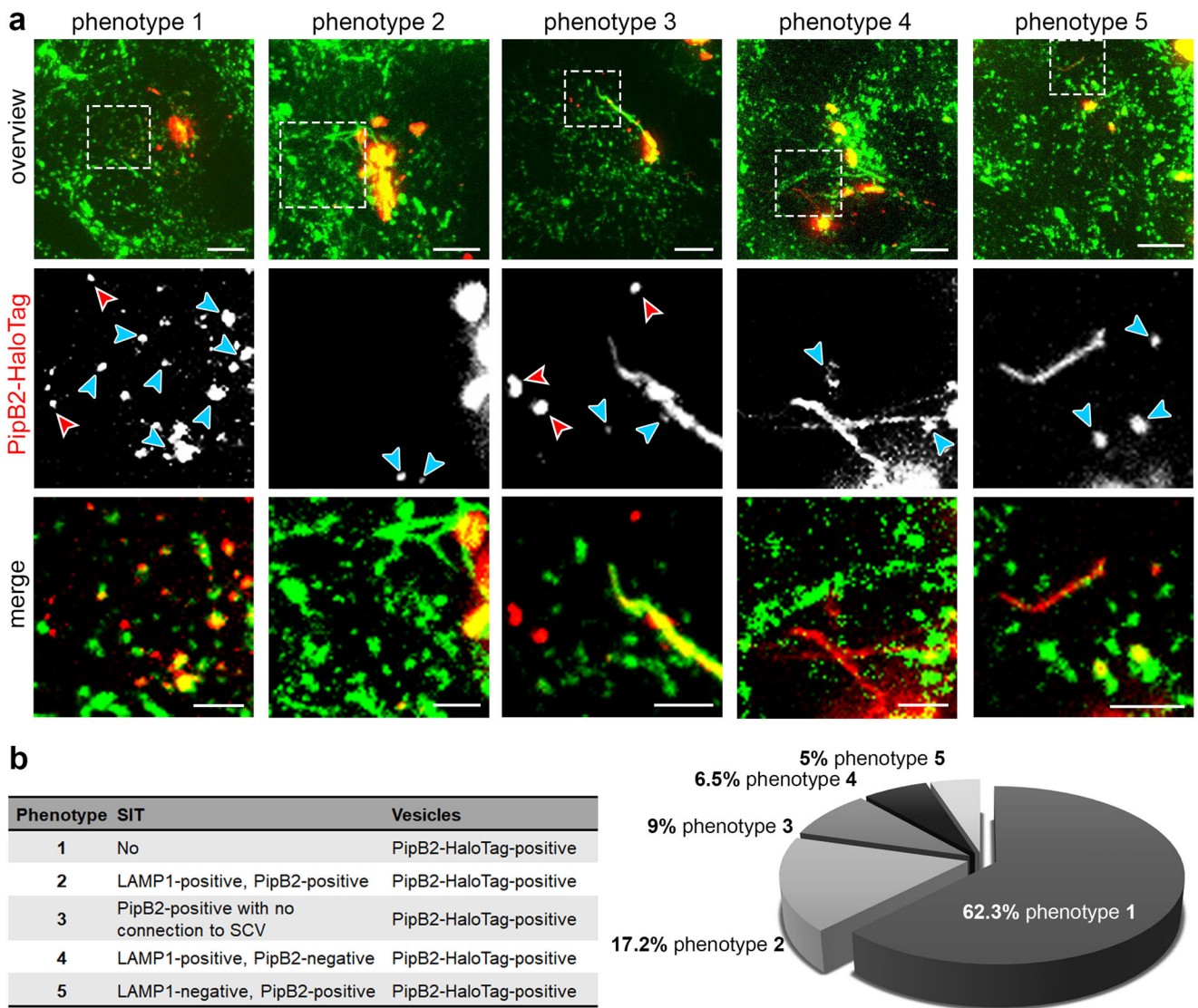

**Fig. 6 | Distribution of PipB2-HaloTag in the early phase of infection.** HeLa LAMP1-GFP cells were infected with STM *pipB2* mutant strain expressing *pipB2*::-HaloTag::HA. **a** LCI was performed directly after cells were stained with 1 µM HTL-TMR at 3.5 h p.i. for 30 min. For infected cells with PipB2-HaloTag-positive vesicles, the phenotypes of PipB2-HaloTag localization on vesicles and SIT were determined. Blue arrowheads indicate vesicles double-positive for LAMP1-GFP and HTL-TMR-

labeled PipB2-HaloTag. Red arrowheads indicate vesicles negative for LAMP1-GFP and positive for PipB2-HaloTag. At least 100 infected cells from four independent experiments were analyzed. Scale bars: 10 and 2 µm in overviews and details, respectively. **b** Quantification of distinct PipB2-HaloTag distributions in infected HeLa LAMP1-GFP cells.

Interestingly, the mobility of effectors differs with PipB2 being the most mobile effector protein. Mobility of membrane proteins depends on membrane integration, membrane composition and interaction with other proteins. The association of PipB2 to lipid rafts has been demonstrated[18]. These dense membrane patches would presume an impaired mobility for membrane-integral proteins. Contrary, it is established that raft association is not the dominant factor in determining the overall mobility of a particular protein as different lipid raft associated proteins showed diverse DC[32]. Besides this, our experiments did not reveal a specific localization of PipB2 to distinct regions of SIF membranes.

Correlation of different forms of protein association with membranes to the corresponding diffusion mobility revealed that DC of prenylated proteins were higher than DC of transmembrane proteins[32]. We did not observe such a correlation for STM effector proteins, with SifA being the only prenylated effector protein tested[33], and propose that mobility of effectors might be influenced to a large extend by their interaction with host proteins. SseF interacts with STM effector SseG

and in combination with the Golgi network-associated protein ACBD3, forming a complex tethering the SCV to the Golgi[34]. SifA has various interaction partners inside the host cell, such as PLEKHM2 (also called SKIP) and PLEKHM1[22,23]. The SifA complex is able to activate kinesin-1 upon binding, resulting in budding and anterograde tubulation of SIF membranes along microtubules[3,35,36]. PipB2 is responsible of the recruitment of auto-inhibited kinesin-1 to the vacuole membrane[37]. The distinct DC of effector proteins SifA, SseF and PipB2 likely indicates the form of interaction with cognate host cell molecules. Both, SifA and SseF, take part in tethering SCV and SIF to host structures, possibly resulting in reduced mobility. Hence, the molecular mechanisms behind effector mobility on SIF membrane remain to be elucidated.

We demonstrate here that effector motilities are affected by SIF architecture. The reduced DC of STM effector protein SifA, as well as host protein LAMP1 reveals slower movement on sm SIF. This indicates an overall impediment of mobility of membrane proteins in the membrane of sm SIF. This may be due to the lower membrane area of

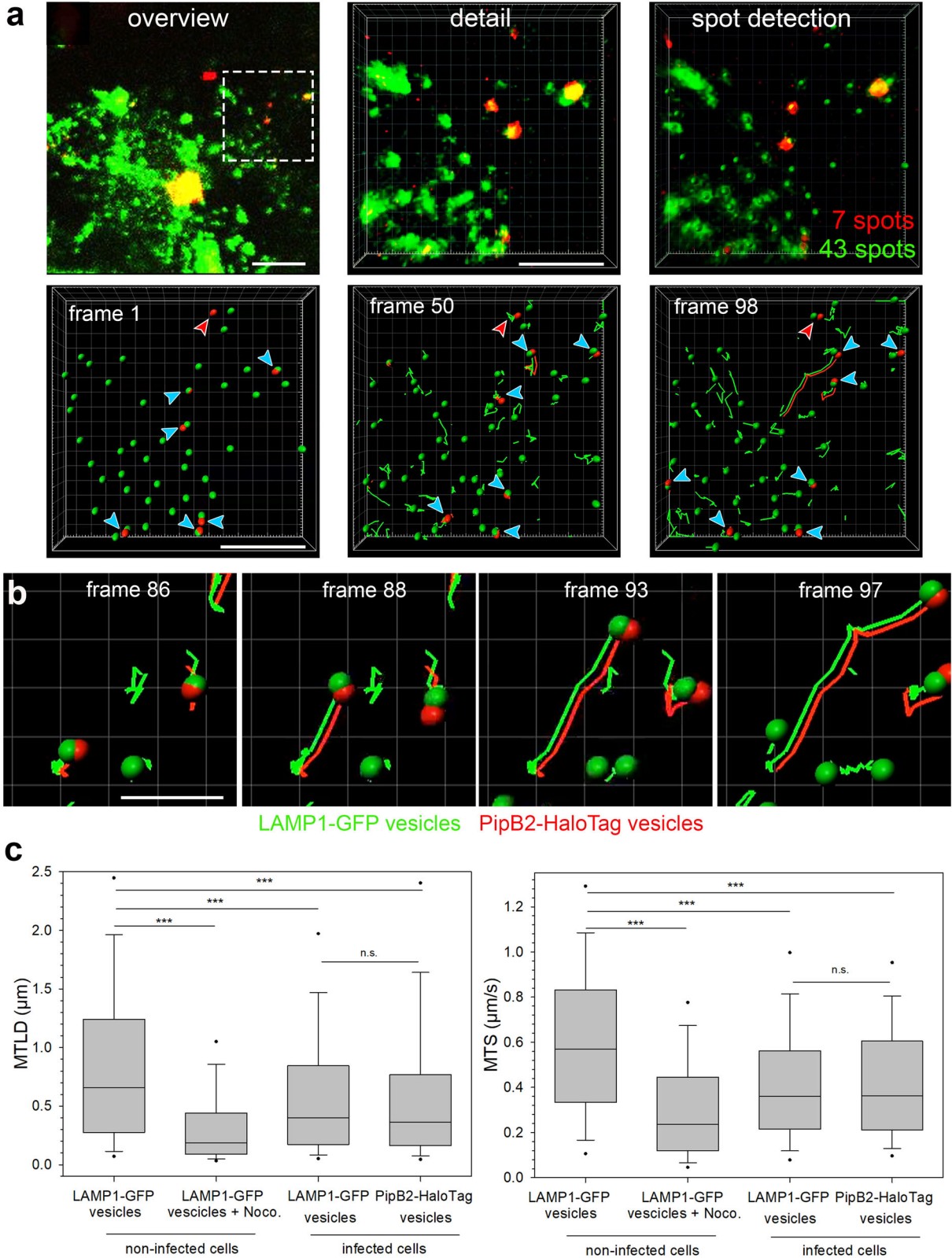

LAMP1-GFP vesicles    PipB2-HaloTag vesicles

sm SIF that limits diffusion of membrane proteins perpendicular to the longitudinal axis of SIF. The smaller diameter of sm SIF compared to dm SIF leads to higher membrane curvature of sm SIF, and this may affect motility of membrane proteins. Furthermore, host cell proteins sensing membrane curvature will be recruited differentially to sm SIF and dm SIF depending on their curvature, and these proteins could have distinct effects on motilities of the proteins analyzed.

When monitoring growing SIF, we found thinner leading SIF at the extending tips, followed by development of thicker trailing SIF. Prior ultrastructural analysis revealed that leading SIF are single-membrane tubules, while trailing SIF comprise double-membrane structure, representing the fully developed SIF. It was proposed that effector protein SseF together with its interaction partner SseG are involved in conversion of sm SIF to dm SIF[5]. Here we present, to our knowledge

**Fig. 7 | Tracking of vesicles positive for LAMP1-GFP and PipB2-HaloTag.** HeLa LAMP1-GFP cells were either not treated, treated with nocodazole to inhibit vesicle movement, or infected with STM Δ*pipB2* strain expressing *pipB2*::HaloTag::HA. **a** An infected HeLa LAMP1-GFP cell with LAMP1-GFP (green) and PipB2-HaloTag-TMR (red) was imaged for 200 frames (0.39 frames/sec) by SDM in dual camera streaming mode. Vesicle tracking analysis was done with the Imaris spot detection tool and co-motion analysis is shown at different time points (Supplementary Movie 12). Blue arrowheads indicate vesicles positive for LAMP1-GFP and effector protein fused to HaloTag and labeled with TMR. Red arrowheads indicate vesicles negative for

LAMP1-GFP and positive for effector protein. **b** Trajectories of single vesicles labeled with LAMP1-GFP and PipB2-HaloTag. Scale bars: 10 and 5 μm in overviews and details, respectively in **a**, 2 μm in **b**. **c** Quantification of at least 858 trajectories from five individual cells per condition. Box plot analysis of mean track displacement length (MTDL) and mean track speed (MTS) of vesicles under various conditions. Boxes indicate 25th and 75th percentiles, the lines within boxes mark medians, whiskers above and below boxes indicate the 90th and 10th percentiles, and dots indicate outliers. Statistical analyses were performed by two-sided Rank Sum test and significances are indicated as follows: n.s., not significant, *$p < 0.05$, ***$p < 0.001$.

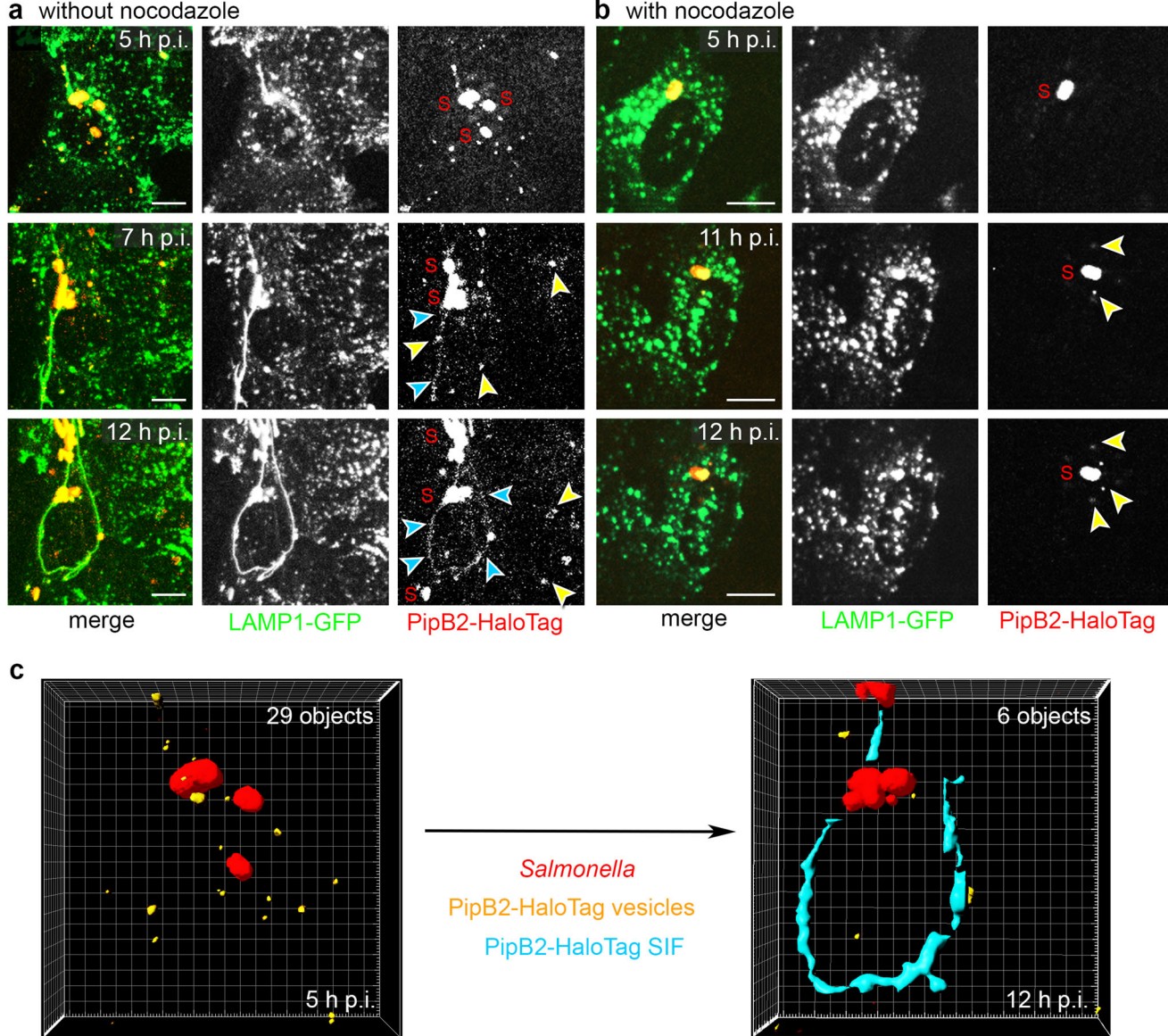

**Fig. 8 | Conversion of vesicular to tubular distribution of translocated effector proteins.** HeLa LAMP1-GFP cells were infected with STM Δ*pipB2* strain expressing *pipB2*::HaloTag::HA. Cells were either not treated (**a**), or treated with nocodazole (5 μg ml⁻¹) 2 h p.i. **b** The inhibitor was removed after HaloTag staining and cells were washed twice. LCI was performed using SDM directly after cells were stained with 1 μM HTL-TMR for 30 min. The cells were imaged over a period of 8 h every 30 min (Supplementary Movie 15, Supplementary Movie 16). Representative STM are

labeled (S), and PipB2-HaloTag-positive vesicles or SIF are indicated by yellow or blue arrowheads, respectively. Micrographs representative for infected cells at indicated time points from three independent experiments are shown in **a**, **b**. Scale bars: 5 μm. **c** The Imaris surface analysis tool was used to determine in an infected cell at 5 h and 12 h p.i. the amounts of either vesicles (orange), or SIF tubular structures (blue) positive for PipB2-HaloTag. The analysis was performed for events in a cell representative of infected cells in three independent experiments.

first, microscopy-based evidence that effector protein SseF is present on the membranes of leading sm SIF and that the effector accumulates before engulfment to trailing dm SIF. As this is also true for SifA and PipB2, more effector proteins might be involved in the process of sm SIF to dm SIF conversion.

The concentration of effectors on the tips of growing SIF led to the question how STM effectors are able to accumulate and how they are recruited after secretion into the host cytosol. Especially SseF which is characterized by large hydrophobic domains[14], could be mistargeted to cytoplasmic membranes or be prone to aggregation in

host cytosol. To test models for the delivery of effector proteins to SCV-SIF continuum introduced in Fig. 3, we performed LCI with labeled effectors in the early infection. We found that various SPI2-T3SS effectors decorate host endomembranes in infected cells. The association of PipB2 with vesicles in the periphery of infected host cells has already been demonstrated[18]. Over time, the presence of PipB2-positive vesicles declined, corresponding with the formation of PipB2-positive SIF. These findings strengthen the hypothesis of delivery of effector proteins to SCV and SIF via fusion of effector-containing endomembranes as depicted in models **b** and **c** of Fig. 3.

The targeting of bacterial effector proteins to phagosomes and various organelles in infected host cells has been widely demonstrated[38], but only few studies observed effector protein presence on host endomembranes and connected a trafficking mechanism of effector proteins to these phenomena. The *Shigella* effector protein IpgB1 was found to localize to endocytic vesicles in mammalian cells expressing eGFP-tagged IpgB1. Vesicles decorated with IpgB1 were found to be functional in host cell trafficking and a relocation of effector proteins from endocytic vesicles to membrane ruffles produced by *Shigella* was observed, indicating a delivery of effector via vesicle fusion[39]. Another study uses the split-GFP approach to monitor the delivery of effector AvrB by *Pseudomonas* via the T3SS to infected plant cells. In infected host cells, AvrB localizes to the plasma membrane, but at different time points after inoculation the localization changed to unknown vesicles, suggesting a potential trafficking of AvrB on vesicles[40]. Relocation of bacterial effector proteins on host endocytic vesicles to the required site of action in infected host cells might represent a universal delivery mechanism.

To further address the question how effector proteins reach their subcellular destination on host endomembranes, we analyzed effector protein localization in infected cells with inhibited vesicle movement. We found that endomembranes did not acquire PipB2 when vesicle movement was inhibited. This finding supports an insertion route of effector proteins to membranes that requires vesicle movement. In this case, effector proteins that do not require a modification by host proteins are inserted into vesicle membranes in close proximity to the T3SS, either by host chaperones, or solely determined by the sequence of membrane-integral effectors (Fig, 3, model **c**).

However, these hypotheses can only be applied to effector proteins without the requirement for host cell modification. SPI2-T3SS effector proteins can gain membrane association either by defined transmembrane domains within their sequence, or by host cell modifications. The effector proteins SseF and SseG are examples for the former, as these proteins contain identified transmembrane domains[8,41,42]. Various other SPI2-T3SS effector proteins acquire membrane association by host cell modifications. These modifications comprise different mechanisms to increase the overall hydrophobicity of effector proteins. SifA is targeted by host modifications, resulting in S-prenylation of the effector protein[33]. Additionally, the effector proteins SspH2 and SseI of STM have been shown to be palmitoylated to gain membrane association[43]. As the mentioned models do not include effectors requiring post-translational modification by the host, other mechanisms must contribute to effector relocation (Fig, 3, model **d**). Accordingly, we did not detect SifA on host endomembranes in infected cells.

Our data lead to various new questions. (i) During the intracellular life of STM, host endosomal compartments are gradually depleted due to fusion to SCV-SIF continuum[44]. Does this terminate the delivery of effector proteins and end intracellular proliferation? (ii) By which mechanism are post-translationally modified effector proteins relocated? iii) Are endomembranes specifically targeted to the T3SS, or is insertion simply dependent on proximity? Further LCI and single molecule-based analyses of STM effectors will likely contribute to answer these questions.

## Methods

### Bacterial strains and culture conditions

Infection experiments were performed using *Salmonella enterica* serovar Typhimurium (STM) NCTC 12023 strain as WT and isogenic mutant strains (Table 1). Mutagenesis was carried out as described elsewhere[45]. In short, strains were constructed using λ Red-mediated mutagenesis and the resistance cassette was removed using FLP-mediated recombination. Mutant strains deficient in effector genes harboring plasmids encoding the corresponding effector fused to HaloTag (Table 2) were used for microscopic analysis.

### Generation of vectors for effector protein-SLE fusions and mutant alleles

Plasmids were constructed as described previously[16] using oligonucleotides listed in Supplementary Table 1. Bacterial strains were cultured in Luria-Bertani broth (LB) containing $50\,\mu g\,ml^{-1}$ Carbenicillin (Roth, 6344.3).

For generation of a plasmid encoding triple HA-tagged SseF, p2643 (*sscB sseF*::HA) was used and *sseF*::HA on p2643 was replaced by *sseF*::3HA using Gibson assembly GA. Primers for generation of vector fragment, check primers and sequence of synthetic *sseF*::3xHA (gBlocks, IDT) are listed in Supplementary Table 1.

**Table 1 | *Salmonella enterica* serovar Typhimurium strains used in this study**

| Designation | Relevant characteristics | Reference |
|---|---|---|
| NCTC 12023 | Wild type | Lab stock |
| MvP388 | Δ*sscB sseF*::FRT | [52] |
| MvP392 | Δ*sseJ*::FRT | [53] |
| MvP503 | Δ*sifA*::FRT | [54] |
| MvP742 | Δ*steC*::FRT | This study |
| MvP1900 | Δ*sseJ*::aph Δ*sifA*::FRT | [45] |
| MvP1944 | Δ*pipB2*::FRT | [44] |
| MvP1948 | Δ*sseF*::aph Δ*sifA*::FRT | This study |
| MvP1980 | Δ*sseF*::FRT | [55] |

**Table 2 | Plasmids used in this study**

| Designation | Relevant genotype | Source/reference |
|---|---|---|
| p2095 | P$_{sseA}$::*sscB sseF*::M45 | [52] |
| p2129 | P$_{sseJ}$ *sseJ*::M45 | [41] |
| p2621 | P$_{pipB2}$ *pipB2*::M45 | [18] |
| p2643 | P$_{sseA}$::*sscB sseF*::HA | [52] |
| p2888 | P$_{sseA}$::*sscB sseF*::HA *sseG*::M45 | [52] |
| p3805 | P$_{CMV}$::LAMP1-miniSOG-mCherry | this study |
| p3806 | P$_{CMV}$::LAMP1-tdminiSOG-mCherry | this study |
| p3991 | LAMP1::HaloTag::GFP | [9] |
| p4305 | P$_{sifA}$ *sifA*::L16::HaloTag::HA | [16] |
| p4118 | P$_{sseA}$ *sscB sseF*::L16::HaloTag::HA | [16] |
| p4286 | P$_{sseJ}$ *sseJ*::L16::HaloTag::HA | [16] |
| p4295 | P$_{pipB2}$ *pipB2*::L16::HaloTag::HA | [16] |
| p5059 | P$_{steC}$ *steC*::L16::HaloTag::HA | This study |
| p5065 | P$_{sseA}$::*sscB sseF*::3HA | This study |
| p6172 | P$_{pipB2}$ *pipB2* ΔCC::L16::HaloTag::HA | this study |
| p6173 | P$_{sopD2}$ *sopD2* ΔCC::L16::HaloTag::HA | this study |
| p6187 | P$_{sifA}$ *sifA* Δ331-336::L16::HaloTag::HA | this study |
| p6203 | P$_{sseJ}$ *sseJ* ΔCC::L16::HaloTag::HA | this study |

Mutagenesis was performed for plasmids p4286, p4295, p4300, and p4305 encoding *sseJ*::HaloTag, *pipB2*::HaloTag, *sopD2*::HaloTag or *sifA*::HaloTag, respectively, using primers listed in Supplementary Table 1. Deletion of codons 331-336 in *sifA* removes the previously described prenylation site[24] and was expected to reduce membrane interaction. Mutations Y281D Y284D K288D (*sopD2*), N309D L312D V316D (*sseJ*), and Y54D L57D M61D (*pipB2*) affect coiled-coil (CC) domains that were reported to contribute to membrane interactions of translocated T3SS effector proteins[25]. Site-directed mutagenesis was performed using the Q5 SDM kit (NEB, E0554S), and resulting plasmids were confirmed by sequencing (SeqLab, Göttingen).

## Culture of eukaryotic cells

The non-polarized epithelial HeLa cell line (ATCC no. CCL-2) stably transfected with LAMP1-GFP[5] was cultured in Dulbecco´s modified Eagle´s medium (DMEM) containing $4.5 \, g \times l^{-1}$, glucose, 4 mM stable glutamine and sodium pyruvate (Merck Sigma-Aldrich, D6429) and supplemented with 10% inactivated fetal calf serum (Thermo, Gibco, 10270). Cells were maintained in a cell culture incubator (37 °C, 5% $CO_2$). Murine macrophage-like cell line RAW264.7 stably transfected with LAMP1-GFP cells was cultured as described before[5].

## Host cell infection

HeLa LAMP1-GFP cells were seeded in surface-treated eight-well plates ((Ibidi, 80827) or on 24 mm diameter coverslips (VWR, Marienfeld 0117640) in six-well plates (Faust, TPP92006). For infection experiments, cells were grown to 80% confluence (eight-well: ~$8 \times 10^4$, six-well: ~$6 \times 10^5$). STM strains were grown overnight in LB supplemented with $50 \, \mu g \, ml^{-1}$ carbenicillin and subcultured 1:31 in fresh LB for 3.5 h. Cells were infected at MOI 50 or 75 for 25 min, washed thrice with phosphate-buffered saline (PBS), and incubated with DMEM containing $100 \, \mu g \, ml^{-1}$ gentamicin (Applichem, A1492) to kill non-internalized bacteria. Subsequently, medium was replaced by DMEM containing $10 \, \mu g \, ml^{-1}$ gentamicin for the rest of the experiment.

RAW264.7 LAMP1-GFP cells were seeded in surface-treated 8-well plates. For infection experiments, cells were grown to 80% confluence (per well ~$1.6 \times 10^5$ cells). RAW264.7 cells were activated with $5 \, ng \, ml^{-1}$ recombinant murine Interferon-γ (R&D system, 485-MI) 1-day prior infection. For infection of RAW264.7 cells, STM strains were grown overnight in LB supplemented with $50 \, \mu g \, ml^{-1}$ carbenicillin. Cells were infected at MOI 50 for 25 min, washed thrice with phosphate-buffered saline (PBS), and incubated with DMEM containing $100 \, \mu g \, ml^{-1}$ gentamicin to kill non-internalized bacteria. Subsequently, the medium was replaced by DMEM containing $10 \, \mu g \, ml^{-1}$ gentamicin for the rest of the experiment (11 h).

## Pulse-chase with gold nanoparticles

Pulse-chase labeling of the endosomal system with custom-made rhodamine-conjugated gold nanoparticles was performed as previously described[17].

## Correlative light and electron microscopy (CLEM)

CLEM of HeLa LAMP1-GFP cells infected by STM WT was performed as previously described[5]. Briefly, HeLa LAMP1-GFP cells were grown on MatTek dishes with a gridded coverslip (MatTek P35G-1.5-14-CGRD-D) and infected with the respective STM strains at MOI 75. Cells were fixed with 2% paraformaldehyde (PFA, Sigma-Aldrich, 158127) and 0.2% glutaraldehyde (GA, Sigma-Aldrich 111-30-8) in 0.2 M HEPES (Sigma-Aldrich, H7006) for 30 min prior to LM. After rinsing the cells for three times with 0.2 M HEPES buffer, unreacted aldehydes were blocked by incubation with 50 mM glycine in buffer for 15 min, followed by rinses in buffer. CLSM was performed and ROIs were chosen. Afterwards cells were fixed with 2.5% glutaraldehyde and 5 mM $CaCl_2$ in 0.2 M HEPES in preparation for TEM. Further steps including post-fixation,

dehydration, sectioning and imaging in the TEM were conducted as previously described[5].

## Live-cell super-resolution localization and tracking microscopy

Localization and tracking of single molecules were done as previously described[16]. Briefly, infected HeLa LAMP1-GFP or RAW264.7 cells were labeled with the HaloTag ligand coupled to tetramethylrhodamine (HTL-TMR, Promega G8251) in a concentration of 20 nM for 15 min at 37 °C. After 10 washing steps, the cells were imaged with an imaging medium consisting of Minimal Essential Medium (MEM) with Earle's salts, without $NaHCO_3$, without L-Glutamine, without phenol red (Merck D1145) and supplemented with 30 mM HEPES, pH 7.4. TIRF microscopy was performed using an inverted microscope Olympus IX-81, equipped with an incubation chamber maintaining 37 °C and humidity, a motorized 4-line TIRF condenser, a ×150 objective (UAPON 150x TIRF, NA 1.45), a TIRF quadband polychroic mirror (zt405/488/561/640rpc), a 488 nm laser (150 mW, Olympus), and a 561 nm laser (150 mW, Olympus). Localization, as well as tracking of single molecules, were carried out with the help of a self-written user interface in MatLab 2013a[21,46–50].

## Live-cell fluorescence microscopy

Infected HeLa LAMP1-GFP cells were labeled with HTL-TMR with a final concentration of 1 μM for 30 min directly before LCI. Cells were washed thrice with PBS and the media was replaced with imaging media as described above. For inhibition of vesicle movement cells were incubated with $5 \, \mu g \, ml^{-1}$ nocodazole (Sigma-Aldrich, M1404) 2 h p.i. Directly before LCI, cells were washed twice to remove nocodazole and labeled as described above. Imaging was done using either the confocal laser-scanning microscope Leica SP5 (Leica) equipped with an incubation chamber, a ×100 objective (HCX PL APO CS, NA 1.4-0.7) and a polychroic mirror (TD 488/543/633) or the Cell Observer Spinning Disk microscope (SDM, Zeiss) equipped with a Yokogawa Spinning Disc Unit CSU-X1a 5000, an incubation chamber, ×63 objective (α-Plan-Apochromat, NA 1.4), two EMCCD cameras (Evolve, Photometrics) and the filter combination GFP with BP 525/50 and RFP with BP 593/46 for dual cam imaging. Images were acquired and processed using the corresponding software LAS AF (Leica) and ZEN (Zeiss).

## Image analysis by Imaris

Vesicle tracking and surface analysis were analyzed by Imaris 9.2.1 software package (Bitplane). Surface analysis was done to determine the amounts of effector-positive vesicles inside an infected cell. By the surface tool, data acquired by LCI were analyzed in the red (PipB2-HaloTag-TMR) channel. Vesicle volume was adjusted using auto-threshold and smoothing of 0.1, and SIF volume was adjusted using auto-threshold and smoothing of 0.3. For vesicle tracking, the spot tool was used. Spot detection was performed with the following parameters XY diameter: 0.75 μm, active model PSF elongation: 1.5 μm, and background subtraction. For tracking the autoregressive motion algorithm was chosen with a maximum gap size of 3.

## Image analysis by FIJI

The FIJI package[51] was used to determine intensity profiles of accumulated SifA-HaloTag trajectories on SIF. Using the line tool and subsequently, the plot profile tool, the intensity profile of the trajectories labeling the SIF was calculated and the relative distances in pixel were compared.

## Statistics and reproducibility

Each experiment was repeated at least twice with similar results, using independent experimental samples and statistical tests as specified in the figure legends. Source data with sample sizes, number of technical and/or biological replicates, means, standard

deviations, and calculated *p* values (where applicable) are provided in the Source Data file for Figures and Supplementary Figures.

## Reporting summary

Further information on research design is available in the Nature Portfolio Reporting Summary linked to this article.

## Data availability

Source data are provided with this paper. The microscopy data generated in this study have been deposited in the OsnaData repository under accession code: https://doi.org/10.26249/FK2/DC45CV. All plasmids described in this work are available at Addgene (Addgene IDs 198142-198151) Source data are provided with this paper.

## Code availability

The code for the SlimFast application is available under: https://github.com/CPaoloR/SLIMfast.

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

## Acknowledgements

This work was supported by the DFG through grants HE 1964/18-2, SFB 994 project Z, and SFB 1557 project 8 to M.H. We like to thank Jacob Piehler (Div. Biophysics), Christian Richter (Div. Biophysics), Rainer Kurre (iBiOs), and Michael Holtmannspötter (iBiOs) for continuous support and fruitful discussions. The support by Susanne Kunis in data management is kindly acknowledged.

## Author contributions

Conceptualization, V.G., M.H.; methodology, V.G., N.S., M.S., F.S., R.F., V.L., O.E.P., and M.H., investigation, V.G., N.S., M.S., F.S., R.F., V.L., O.E.P., and M.H.; formal analysis, V.G., N.S., M.S., F.S., R.F., V.L., O.E.P., and M.H.; validation, V.G., N.S., V.L., O.E.P., and M.H.; writing, V.G., M.H.; supervision, O.E.P., M.H.; funding acquisition, M.H.

## Funding

## Competing interests

The authors declare no competing interests.
