## [Peer Review File · Nature Communications]

Single molecule analyses reveal dynamics of Salmonella translocated effector proteins in host cell endomembranesREVIEWER COMMENTS

Reviewer #1 (Remarks to the Author):

The aim of this well written paper was to study the route a selected STM SPI2-T3SS effectors use to disseminate into host endosomal membranes and the expanding SIF network. Using contemporary imaging and tagging techniques the authors followed intracellular effector trafficking inside HeLa cells expressing LAMP1-GFP.

The study is well constructed, technically solid, and critically assessed. The proposed dissemination models presented in Fig. 3 are rationale and form a good basis for the experimental design.

The authors uniquely show that SPI2-T3SS effector proteins are recruited to vesicles during early infection, which are then fused with the SCV-SIF continuum.

Overall, the paper provides novel advances in our understanding of the STM intracellular effector trafficking. Nonetheless, I believe few gaps remain.

Major comments

1. The authors describe subcellular distribution of SseF (as well as PipB2 and SseJ) over time. Since SseF binds SseG the authors should investigate if trafficking of SseF is dependent on SseG (using an sseG mutant) and where/when SseF and SseG form a complex (i.e. is it in the vesicles or within the SCV-SIF continuum).

2. The data are based entirely on visualization techniques and as such as missing some negative controls/quantification. The study would benefit from inclusion of specific effector mutants (e.g. lacking membrane targeting domains) that could be used to show specificity and be used for validation/quantification.

3. HeLa cells, while providing an excellent first model, do not represent the physiological Salmonella target cells. The authors should validate some of the key conclusions in mouse and human macrophages (particularly as they state that differential involvement of multiple host proteins implicated in intracellular transport may explain cell line-dependent variations in the pathophysiology of Salmonella infections).

Reviewer #2 (Remarks to the Author):

Göser et al. report investigations of translocated effector proteins of Salmonella-induced filaments (SIF). The authors made use of CLEM imaging to identify LAMP1-positive SIF in infected cells, which was confocal imaging in fixed cells, followed by post-fixation and classical TEM. It's an appropriate methods for the given task. To study the 2D dynamics of the translocated effector proteins in living cells, single molecule tracking was used. From the technological point of view, this has been described already in Göser et al. 2019 (<https://doi.org/10.1128/mBio.00769-19>) for the same biological system (I far as I can judge this). For example, compare Fig. 4 in the manuscript with Fig. 6 in Göser et al. 2019. The only critical point that I see from a technical perspective is that I could neither find the code for single molecule tracking or a link to a repository in the manuscript nor in Göser et al. 2019. In general, the imaging methods used by the authors appear to be appropriate choices, technically sound and the single molecule tracking quite state-of-the-art. As my background is rather technical, I cannot assess the biological content of the manuscript.

Reviewer #3 (Remarks to the Author):

Göser et al. used cutting-edge imaging techniques that they had developed in previous studies to follow effector trafficking in Salmonella-infected epithelial cells. They observed different localization patterns at early vs. late stages of infection, different diffusion rates in thin sm SIFs and large dm SIFs that also varied among different effectors, and detected an accumulation of effectors at the tip of growing SIFs. Some of the findings have already been reported previously, but this study still provides interesting new insights into SIF development. As the authors correctly point out, several key questions can still not be answered conclusively but this is reasonable given how challenging they are.

I have a few comments:

Are data and conclusions derived from a single infected cell for the experiments shown in Fig. 1, 2, 5E, 8, or were similar data observed in several independent biological replicates examining multiple cells? This is particularly important for data that would support key findings of this study including the re-localization of vesicle-associated effectors to SIFs with progressive infection and the accumulation of effectors at the tip of growing SIFs.

Lines 134-6, 153-4: What is the evidence for initial incorporation into vesicles of the endosomal system as opposed to other endomembranes? The detail pictures in Fig. 2A and Fig. 6A for 4h post-infection actually show often limited overlap between green and red channels? This is different on the pictures shown in Fig. S5. The EM data in Fig. 2B show some vesicles at 8h, but it is unclear what they are?

Lines 189-92: Please mention that similar values have already been observed in your previous study for fixed and live cells (LAMP1, SseF)(Göser et al. 2019). There is a discrepancy between values for LAMP1 in the text and in Fig. 4B (0.11 vs. 0.055). Your previous study showed 0.055.

Lines 201-2: Does this suggest that the proteins are using predominantly the outer membrane in dm SIFs?

Line 289: refers rather to Fig. S6

Lines 308-9: this has already been shown in Göser et al. 2019

Lines 341-347: Could the larger area of dm SIF also give merely the impression of more rapid movement because the area perpendicular to the long axis is larger, thus giving additional freedoms of movements? In other words, if a protein would move in circles around a thin sm SIF it would show rather little distances in the xy plane. By contrast, a protein moving in circles around the larger dmSIF would show as more impressive xy movement?

Fig. 2A: the dashed boxes in the overview are a bit confusing as they correspond to either SCV or SIF details

Fig. 3, model C: Perhaps it would be more likely that directly inserted effectors in the SCV membrane can leave the SCV by budding-off of vesicles? Have you ever seen vesicles close to SCV receiving effectors?

Authors' response: We thank the reviewers for the constructive comments on our initial manuscript. The comments have stimulated new experiments, which some time to execute and analyze. The new data are added to the revised version. Some comments have already stimulated further experimental analyses, that we hope to report on in the future.

Please find below a point-by-point response to the comments with our response indicated in blue font.

REVIEWER COMMENTS

Reviewer #1 (Remarks to the Author):

The aim of this well written paper was to study the route a selected STM SPI2-T3SS effectors use to disseminate into host endosomal membranes and the expanding SIF network. Using contemporary imaging and tagging techniques the authors followed intracellular effector trafficking inside HeLa cells expressing LAMP1-GFP.

The study is well constructed, technically solid, and critically assessed. The proposed dissemination models presented in Fig. 3 are rationale and form a good basis for the experimental design.

The authors uniquely show that SPI2-T3SS effector proteins are recruited to vesicles during early infection, which are then fused with the SCV-SIF continuum.

Overall, the paper provides novel advances in our understanding of the STM intracellular effector trafficking. Nonetheless, I believe few gaps remain.

Major comments

1. The authors describe subcellular distribution of SseF (as well as PipB2 and SseJ) over time. Since SseF binds SseG the authors should investigate if trafficking of SseF is dependent on SseG (using an sseG mutant) and where/when SseF and SseG form a complex (i.e. is it in the vesicles or within the SCV-SIF continuum).

Response: This is an interesting suggestion, but challenging to address experimentally. Complex formation of SseF and SseG in living host cells cannot be visualized by currently available techniques. This would require massive technical improvement, i.e. double labeling, higher resolution. Perhaps co-immunodetection of SseF and SseG may serve as surrogate (see Kuhle et al., 2004). We addressed this interesting question by two lines of experimentation:

- a) Analyses of SseF-HaloTag diffusion in background of SseG-deficient mutant strain
- b) Analyses of spatial distribution of SseF and SseG over time after infection.

We found that the spatial distribution of the both effectors is distinct, and increased colocalization occurs over time of infection. This may indicate distinct initial targeting, and interaction by SseF and SseG may lead to fusion of distinct membrane compartments. Using SMT in living host cells, we observed altered diffusion of SseF-HaloTag in dependence of SseG.

The additional data are shown in new Fig. S 10. Both observations are new and clearly demand further in-depth analyses to unravel the molecular basis.

2. The data are based entirely on visualization techniques and as such as missing some negative controls/quantification. The study would benefit from inclusion of specific effector mutants (e.g. lacking membrane targeting domains) that could be used to show specificity and be used for validation/quantification.

Response: Thank you for the suggestion. Indeed, generating mutant effector proteins that lose membrane association and still allow single molecular imaging is a challenging task. We followed this suggestion generated mutant alleles of SifA, PipB2, SseJ, SopD2 based on prior publications. So far, the membrane targeting sequence best characterized so far is the C-terminal prenylation signal of SifA. For other SPI2-T3SS effectors, coiled-coiled domains have been reported as relevant for targeting, but the evidence is mainly by transfection rather than by bacterial translocation. The mutant forms of effector protein-HaloTag fusions were translocated, but generated reduced localized signals. This is likely in line with reduced membrane association and increased amounts of effector localized in the host cell cytosol. In contrast the wild type forms of effectors with membrane associated localization, the cytosolic mutant forms are likely to be highly diffusible, prohibiting precise localization and single molecule tracking. We made similar observations with SLE-tagged proteins in bacteria, and with SPI1-T3SS effector-SLE fusion in our earlier report. Despite these limitations, we performed SMT analyses of those effector proteins that localized to host cell endomembranes.

Interestingly, we determined altered diffusion rates by SMT for effectors. We anticipate that effector proteins neither associated with SIF membranes nor interacting with host cell target proteins would be freely diffusible in host cytosol, and thus too fast to be analysed by our SMT approach. In turn, the observation of increased diffusion rate of mutant SifA-HaloTag would be in line with sole interaction with target proteins like SKIP and Rab9, but loss of prenylation-mediated interaction with compartment membranes.

3. HeLa cells, while providing an excellent first model, do not represent the physiological Salmonella target cells. The authors should validate some of the key conclusions in mouse and human macrophages (particularly as they state that differential involvement of multiple host proteins implicated in intracellular transport may explain cell line-dependent variations in the pathophysiology of Salmonella infections).

Response: We addressed this comment by analyses of single molecule tracking in infected cells of the murine macrophage cell line RAW264.7. This phagocytic cell line is frequently used to study intracellular lifestyle of Salmonella and other intracellular pathogens. Here we used interferon-gamma activated RAW264.7 expressing LAMP1-GFP. We observed that effector-HaloTag fusion proteins were associated with SIF and showed similar bidirectional movement as observed in HeLa cells. Unfortunately, a LAMP1-GFP expressing human macrophage line was not available to further address this question. It should be mentioned that the imaging technique applied demands rather intensive illumination. This will limit application to cells that are sensitive to photodamage.

The congruence of SMT data in HeLa and RAW cells makes us confident that the observation reported are of broader relevance.

Reviewer #2 (Remarks to the Author):

Göser et al. report investigations of translocated effector proteins of Salmonella-induced filaments

(SIF). The authors made use of CLEM imaging to identify LAMP1-positive SIF in infected cells, which was confocal imaging in fixed cells, followed by post-fixation and classical TEM. It's an appropriate methods for the given task. To study the 2D dynamics of the translocated effector proteins in living cells, single molecule tracking was used. From the technological point of view, this has been described already in Göser et al. 2019 (<https://doi.org/10.1128/mBio.00769-19>) for the same biological system (I far as I can judge this). For example, compare Fig. 4 in the manuscript with Fig. 6 in Göser et al. 2019.

Response: Please appreciate that the prior paper (Göser et al., 2019) reported the methodological aspects of effector-SLE fusions, the various applications and experimental benchmarking. While examples of SMT were shown in the previous work, the systematic analyses were performed in the present work. We also addressed specific new questions, see response to reviewer 1.

Most importantly, experimentally addressing the targeting of effector proteins from translocation to specific subcellular targets in host cells is the main achievement of the present work.

The only critical point that I see from a technical perspective is that I could neither find the code for single molecule tracking or a link to a repository in the manuscript nor in Göser et al. 2019.

Response: That you for this suggestion. We added a link to the code in the Methods section.

In general, the imaging methods used by the authors appear to be appropriate choices, technically sound and the single molecule tracking quite state-of-the-art. As my background is rather technical, I cannot assess the biological content of the manuscript.

Response: Indeed, methodology that allows following effector translocation in living host cells, and for SMT of translocated effector proteins is hardly available. We applied the newly developed approach to test hypotheses for effector proteins targeting within host cells. Based on our results we developed new models how effector proteins are delivered, reach targets and contribute to the dramatic reorganization of the host cell endosomal system observed in Salmonella-infected host cells. We are confident that these results will stimulate discussion, refinement of models, and application of the techniques to other host-pathogen interactions.

Reviewer #3 (Remarks to the Author):

Göser et al. used cutting-edge imaging techniques that they had developed in previous studies to follow effector trafficking in Salmonella-infected epithelial cells. They observed different localization patterns at early vs. late stages of infection, different diffusion rates in thin sm SIFs and large dm SIFs that also varied among different effectors, and detected an accumulation of effectors at the tip of growing SIFs. Some of the findings have already been reported previously, but this study still provides interesting new insights into SIF development. As the authors correctly point out, several key questions can still not be answered conclusively but this is reasonable given how challenging they are.

Response: Thank you for the appreciation of the work. We look forward to address further questions using these techniques.

I have a few comments:

Are data and conclusions derived from a single infected cell for the experiments shown in Fig. 1, 2, 5E, 8, or were similar data observed in several independent biological replicates examining multiple cells? This is particularly important for data that would support key findings of this study including the re-localization of vesicle-associated effectors to SIFs with progressive infection and the accumulation of effectors at the tip of growing SIFs.

Response:

The micrographs shown in Fig.1 are show representative events in of endosomal remodeling in an infected cell. Data shown in Fig 2 were used to demonstrate representative phenotypes and not quantified. We extended the presentation of ultrastructural data by showing further events, and samples processed by distinct preparation techniques.

Data shown in Fig. 5 were quantified from 20 or more infected cells per conditions, resulting from independent experiments. This data set led to the conclusion of effector accumulation at tips pf growing SIF.

The data shown in Fig. 8 are representative for several independent experiment. However, the full quantification as shown was performed for a representative example.

Lines 134-6, 153-4: What is the evidence for initial incorporation into vesicles of the endosomal system as opposed to other endomembranes? The detail pictures in Fig. 2A and Fig. 6A for 4h post-infection actually show often limited overlap between green and red channels? This is different on the pictures shown in Fig. S5. The EM data in Fig. 2B show some vesicles at 8h, but it is unclear what they are?

Response: Indeed, the effector proteins may be inserted in a variety of host cell endomembranes. Our previous proteomic analyses indicated a rather heterogeneous origin of host cell membranes recruited to SIF and SCV.

Indeed, a conclusion we draw from the distribution of effector protein shown in Fig. 2A is that initial localization is distinct from SCV/SIF membrane, and that association of effector proteins with LAMP1-positive SCV/SIF membranes increases for time of intracellular activity of Salmonella. This observation also applies to other effectors shown in Fig. S 3, and supported by the new data shown in Fig. S 10. Data in Fig. S 10 also indicate that dynamics of association of effectors with SIF/SCV for specific effectors.

The immuno-EM in Fig. 2B was performed using Tokoyasu technique to identify effector decorated vesicles. As these vesicles are spherical and not part of a tubular network, we propose that these vesicles were targeted by effectors. However, the precise organellar identity cannot determined by this technique.

Lines 189-92: Please mention that similar values have already been observed in your previous study for fixed and live cells (LAMP1, SseF)(Göser et al. 2019). There is a discrepancy between values for LAMP1 in the text and in Fig. 4B (0.11 vs. 0.055). Your previous study showed 0.055.

Response: Thank you for pointing out the mistake. The value was corrected to match the figure. Reference to prior data was added.

Lines 201-2: Does this suggest that the proteins are using predominantly the outer membrane in dm SIFs?

Response: This is a good point. So far, no imaging techniques is available to resolve the position of effector at inner of outer membrane of SIF. This would require labeling and detection in the low nanometer range. We are currently exploring Nanobodies for this question, but a label suitable to TEM is still pending.

If we propose SIF formation by lateral extension of membrane of single membrane tubules, initially effectors present in these membranes would be distributed equally. If dm SIF extend by fusion with further effector-carrying vesicles, the delivery would be primarily to the outer membrane of dm SIF.

Line 289: refers rather to Fig. S6

Response: Yes, we corrected this mistake.

Lines 308-9: this has already been shown in Göser et al. 2019

Response: change text to 'method applied'

Lines 341-347: Could the larger area of dm SIF also give merely the impression of more rapid movement because the area perpendicular to the long axis is larger, thus giving additional freedoms of movements? In other words, if a protein would move in circles around a thin sm SIF it would show rather little distances in the xy plane. By contrast, a protein moving in circles around the larger dmSIF would show as more impressive xy movement?

Response: Yes, this may indeed affect, likely to a smaller extend, the overall movement. However, the spatial resolution of perpendicular movement on SIF tubules is too low to address this possibility. Please appreciate that the velocity is calculated from the collecting a large number of vectors of molecular migration in a time interval. We added this thought to the discussion (lines 381 ff).

Fig. 2A: the dashed boxes in the overview are a bit confusing as they correspond to either SCV or SIF details

Response: We revised the presentation of clarity and distinguished SCV with boxes with dashed lines from SIF with boxes with dotted lines (Fig. 2A and Fig. S 3).

Fig. 3, model C: Perhaps it would be more likely that directly inserted effectors in the SCV membrane can leave the SCV by budding-off of vesicles? Have you ever seen vesicles close to SCV receiving effectors?

Response: Thank you for this suggestion. We added this possibility as a further option to model A. Our prior work reported that segments of Salmonella-induced filaments (SIF) can bud off a larger tubular structure, move within a cell, and later fuse to other SIF tubules (PMID: 18817527). In these analyses, only a host cell endosomal marker was followed, but very likely effector proteins associated with the membrane show similar behavior.

The directly visual events proposed in model C will be very challenging. We applied SLE labelling after translocation and removed unbound SLE ligands by washing. This process causes a temporal delay, that will prevent to capture the very moment of delivery. I would mention that a fluorogenic SLE substrate has recently been made available. This substrate may allow no wash labelling protocols, and such approach may allow detection of translocated effectors immediately after translocation. We started use of this ligand, but making this working and generated data is hopefully worth a future report.

REVIEWERS' COMMENTS

Reviewer #1 (Remarks to the Author):

The authors have addressed all my comments (or appropriately explained why some suggestions could not be addressed experimentally at this point) and improved what was a very good paper already.

Gad Frankel

Reviewer #3 (Remarks to the Author):

The authors answered adequately to my comments and questions.